# Respiratory infection transmission risk and indoor air quality at outpatient departments and emergency treatment units of Sri Lankan teaching hospitals

N.D.B. Ehelepola 🄳 *, H.M. Arjuna Thilakarathna

Teaching (General) Hospital-Peradeniya, Peradeniya, Sri Lanka

\* drehelepola@gmail.com

**Data Availability Statement:** All relevant data is in the manuscript or in the supplementary file.

## Abstract

Indoor carbon dioxide ($CO_2$) concentration has been used as a proxy of the degree of ventilation and, by extension, as an indicator of the risk of contracting respiratory infections. No publications exist regarding indoor air quality (IAQ) parameters of Sri Lankan hospitals. We measured the levels of $CO_2$ and seven other IAQ parameters during morning rush hours for three days, in outpatient departments (OPDs) and emergency treatment units (ETUs) of all 21 teaching hospitals of Sri Lanka. We measured the same parameters of outdoor air also. We calculated the mean values of those parameters. We looked for correlations between outdoors and OPD and ETU levels of selected air quality parameters. The average $CO_2$ levels of outdoors, OPDs and ETUs respectively were 514ppm (ppm = parts per million), 749ppm and 795ppm. The average levels of $PM_{2.5}$ (particulate matter with diameters <2.5μm) outdoors, OPDs and ETUs respectively, were 28.7μg/m$^3$,32μg/m$^3$ and 25.6 μg/m$^3$. The average levels of $PM_{10}$ (particulate matter with diameters <10μm) outdoors, OPDs and ETUs respectively, were 49.4μg/m$^3$, 55.5μg/m$^3$ and 47.9 μg/m$^3$. The median levels of formaldehyde outdoors, OPDs and ETUs respectively, were 0.03mg/m$^3$, 0.04mg/m$^3$ and 0.08mg/m$^3$. The median levels of total volatile organic compounds (VOC) outdoors, OPDs and ETUs respectively were 0.12mg/m$^3$, 0.19mg/m$^3$ and 0.38mg/m$^3$. $CO_2$ levels of air in OPDs and ETUs generally were below the national ceilings but above the ceilings used by some developed countries. Outdoors, OPDs and ETUs air contain $PM_{10}$, $PM_{2.5}$ levels higher than WHO ceilings, although below the national ceilings. VOC and formaldehyde levels are generally below the national ceilings. Air in OPDs and ETUs is hotter and humid than national ceilings. Outdoor $PM_{10}$, $PM_{2.5}$ levels influence OPDs and ETUs levels. We propose methods to reduce the risk of nosocomial respiratory infections and to improve IAQ of Sri Lankan OPDs and ETUs.

**Funding:** NDBE received support from BALVI. The funders had no role in study design, data collection and analysis, decision to publish, or preparation of the manuscript. Both authors did not receive salary or other funding from commercial companies.

**Competing interests:** The authors have declared that no competing interests exist.

## Introduction

Outpatients and inpatients who seek care at teaching hospitals in Sri Lanka, enter the care pathway via OPDs and ETUs also known as accident and emergency (A&E) units. Studying $CO_2$ levels as a proxy for assessing the risk of respiratory infection transmission at OPDs and ETUs of teaching hospitals in Sri Lanka was the main focus of this study. We studied seven other air quality parameters at the outdoors of hospital premises, indoors of OPDs and ETUs, measured during morning peak hours. Here we present the findings and discuss.

### Current knowledge on how respiratory infections get transmitted and the basis for our use of indoor $CO_2$ levels as a proxy of the risk of contracting respiratory infections

At OPDs and ETUs indoors, people are the predominant source of $CO_2$ as there are no other substantial sources of $CO_2$ like large animals, combustions or decaying organic matter. When coughing, sneezing, talking, and even during normal exhalation, people expel respiratory droplets of diameters of 0.01–1000μm with $CO_2$ [1]. Recent evidence shows that coughing and sneezing (forceful exhalations) create turbulent buoyant puffs containing liquid particles with gas (multiphase) that entrain ambient air and disperse respiratory droplets of up to 8m horizontally depending on factors of the environment [1, 2]. Such droplets exhaled by patients with respiratory infections contain infective microbes. Once exit the nose/mouth, those respiratory droplets progressively shrink in size with evaporation. Larger respiratory droplets fall quicker (in short distances) in semi-ballistic trajectories and a minute fraction of them can get deposited in the respiratory tracts and conjunctivae of the people nearby resulting in infection [2, 3]. Some fall on the surfaces resulting in fomites. But smaller droplets (about <5μm in diameter), known as aerosols, hang in the air for up to a few hours drifting with air currents [2, 4, 5]. When we inhale air containing respiratory droplets/aerosols which comprise infective organisms, we can contract that disease. When there is adequate ventilation, fresh air from outside dilutes and removes the air exhaled by people indoors [6]. Good ventilation brings the indoor $CO_2$ concentrations close to the outdoor level. Global average outdoor $CO_2$ concentration is 417 particles per million = 417ppm and it is rising [7]. High $CO_2$ concentration in the air of patient waiting areas of the OPDs and ETUs compared to the outdoors level indicates the accumulation of air exhaled by people including respiratory droplet aerosols [5, 8, 9]. This happens because ventilation of the room is inadequate to dilute and remove air expired by people inside [5, 8, 9]. Indoor $CO_2$ concentration has been used as a proxy of the degree of ventilation and, by extension, as an indicator of the risk of contracting respiratory infections in that space [5, 8–11]. Until lately, Sri Lankan medical specialists we know and even authoritative organizations like the World Health Organization (WHO) considered only tuberculosis, measles (rubeola) and chickenpox as infections transmitted via aerosols [6, 12]. Nevertheless, recent evidence indicates that many infections including COVID-19 and seasonal influenza that result in high disease burden in Sri Lanka and worldwide are transmitted via aerosols (airborne transmission) [2, 4, 8, 12, 13]. Modeling of the aerosol clouds show that up to 200 virus copies can be deposited in the respiratory tract of a bystander several minutes after coughing /sneezing by a patient [14].

Fig 1 schematically depicts how respiratory infections are transmitted.

COVID-19 pandemic and the availability of non-dispersive infrared (NDIR) sensor based relatively cheap but accurate $CO_2$ monitors have increased the interest and the possibility of wider usage of indoor $CO_2$ concentration as a proxy of the risk of contracting respiratory infections lately [8–11]. Aerosol dispersion in an indoor space is influenced by its geometry and weather parameters [14]. The survival of pathogens in aerosols also depends upon weather

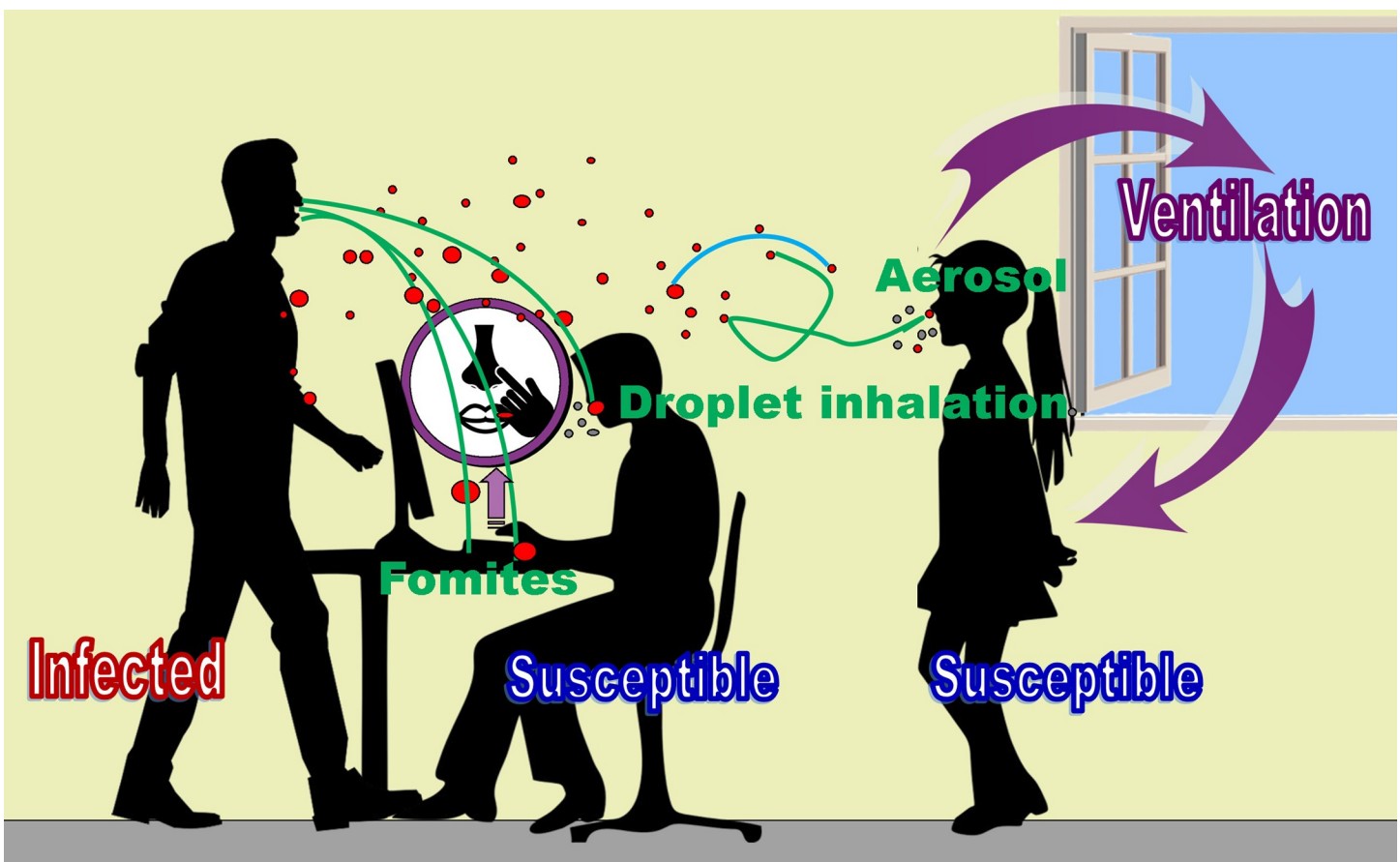

**Fig 1. Schematic representation of how respiratory infections are transmitted from an infected person during a forceful exhalation in indoors.** This illustrates exhalation of infected respiratory droplets of various sizes (red color), how larger ones fall on surfaces resulting fomites, how fomites lead to contraction of infections when touch the face with contaminated hands, how a few droplets can directly get deposited in the respiratory tract and how smaller droplets (aerosols) that hang in the air for a long time infect susceptible people. This depicts how ventilation (outdoor air) dilutes droplet concentration and shrinking of droplet size with evaporation (blue arc) as well. Green droplets are non-infected ones exhaled during normal exhalations. This figure was created by the first author using Microsoft Office2010 software and clipart from https://openclipart.org/.

parameters [13, 14]. Hence, we recorded the temperature and relative humidity (RH%) of our places of study.

### Reasons for our study using $CO_2$ levels as a proxy to assess the risk of contracting respiratory infections at Sri Lankan hospitals

Sri Lanka, is an island nation with 22 million population. Sri Lanka has 22 hospitals categorized as teaching hospitals by the Ministry of Health including two national hospitals (two apex hospitals of the nation) [15]. Waiting areas of the OPDs and ETUs of those hospitals are usually congested especially in the mornings as Sri Lankans generally prefer to go to a larger hospital in a city, lack of mandatory referral/gatekeeping system and because all of them provide services free of charge [16].

At the waiting areas of OPDs and ETUs, usually there are infectious patients infected with respiratory infections and more susceptible people (patients with weaker immune systems due to other diseases) concentrated and mingling with each other. Usually it is not possible to maintain a physical distancing of 1-2m due to the limitations of space. Hence, the risk of

transmission of respiratory infections via aerosols and larger respiratory droplets in those places are higher than in most other public places. There are only a few published studies regarding the risk of nosocomial transmission of respiratory infections in OPDs and ETUs and no such publications are from Sri Lanka [17]. Moreover, high indoor $CO_2$ levels indicate poor indoor air quality (IAQ) and are demonstrated to be associated with sick building syndrome [18].

## An introduction to other IAQ parameters we studied the reasons for including them in our study

High levels of particulate matter and VOC in the air we inhale adversely affect our health as we explain later. People with existing respiratory and cardiovascular diseases, pregnant women and fetuses, elderly and young children are especially vulnerable to such adverse effects [19–22]. Percentages of people in those categories are usually higher in OPDs and ETUs compared to the general population. There are no published studies on following IAQ parameters in hospital settings of Sri Lanka. Therefore, we decided to study air temperature, humidity, $PM_{10}$ and $PM_{2.5}$ levels, total particulate matter levels, VOC and formaldehyde levels as our secondary objective. $PM_{2.5}$, also known as fine particles can remain floating in air for days to weeks. Particles 2.5–10 μm diameter are called coarse particles. Coarse particles remain in the atmosphere for hours [23].

Volatile organic compounds (VOC) are compounds of carbon that evaporate under normal indoor temperature and pressure that can pollute air [24]. Formaldehyde is a common and important VOC that pollutes indoor air [25]. In this study VOC means the total volatile organic compounds detected as VOC by the Temtop LKC-1000S+ second generation professional machine. We measured formaldehyde levels separately.

## How $CO_2$, particulate matter, VOC, indoor air temperature and humidity affect human health

We have explained how high indoor $CO_2$ levels are associated with increased risk of respiratory infections (including COVID-19) transmission. In addition, several studies have demonstrated that $CO_2$ levels > 1000 ppm may be connected to adverse health effects including decreased performance in school or office settings and neurophysiological symptoms like headache, upper respiratory symptoms and eye irritation [8, 26]. One research showed that even a 400ppm rise in the background level is enough to decrease cognitive performance [26]. Bio-effluents including VOCs that rise concomitantly with $CO_2$ in crowded spaces may also contribute to those adverse health effects [26].

Multiple recent studies indicate that $PM_{2.5}$ and $PM_{10}$ levels in the air we breathe are correlated with the risks of COVID-19 and other respiratory infections spread and mortality due to COVID-19 [21, 27]. $PM_{10}$ are also known as ambient particulate matter (PM or $PM_{10}$) [19, 20, 21, 22]. When inhaled, they get deposited in the respiratory tract, particularly in the lung alveoli [19, 21]. $PM_{2.5}$ are particles with diameters less than one-third of the diameter of a red blood cell. $PM_{2.5}$ are even more likely to go deeper in to the lungs down to the alveoli and some of them enter in to the blood in lung capillaries [22, 28]. Those circulate with blood, cross the blood-brain barrier, and cause neurodegenerative disorders and cross the placental and affect the fetuses [22, 28]. Inhalation of PM is known to result and aggravate chronic lung diseases and shown to be increasing the risk of developing cardiovascular diseases, diabetes, some malignancies and neurological disorders etc. as well [19–21, 28, 29].

High levels of VOCs in indoor air can result in irritation of the eyes and upper respiratory tract, headaches, breathing difficulties in the short term, and certain cancers in the long term

[20, 21, 24, 25]. Air temperature and humidity are also known to influence indoor thermal comfort, respiratory infection transmission and influence VOC emission from household items [30–32].

### Known sources in OPDs and ETUs that generate air pollutants we studied

Ambient air contains all those pollutants to a certain level. At indoors of OPDs and ETUs people are the predominant generators of $CO_2$. Particulate matter is generated during nebulization of patients which mostly happens in ETUs, by cleaning agents and cleaning process, from computer printers and fungal spores from damp walls [32]. Settled down particles on floor and other surfaces get re-suspended when people move and linen and equipment are moved [32]. VOCs are generated from cleaning materials including disinfectants, various plastic items, furniture, people and painted surfaces [20, 24, 32].Some VOCs get condensed on indoor surfaces and again get vaporized increasing the duration of exposure [20]. People contribute to rise of temperature and humidity indoors.

## Materials and methods

### Study setting

The 22 hospitals categorized as teaching hospitals by the Ministry of Health of Sri Lanka are the National Hospitals Colombo and Kandy , Teaching Hospital (TH) Peradeniya, TH Karapitiya , Castle Street Hospital for Women , De Soysa Hospital for Women , TH Mahamodara , TH Anuradhapura , TH Colombo North Ragama , TH Batticaloa , TH Jaffna , Lady Ridgeway Hospital for Children , Sirimavo Bandaranaike Specialized Childrens Hospital, TH Colombo South Kalubowila , TH Kuliyapitiya , TH Ratnapura , National Eye Hospital-Colombo, TH (Chest hospital)-Welisara , Rehabilitation Hospital-Ragama, Dental Institute-Colombo , National Institute of Mental Health-Angoda , National Cancer Institute Maharagama (Apeksha), [15]. The largest and most important hospitals of Sri Lanka are among them.

However, the last three out of 22 are sometimes not counted as teaching hospitals by the Ministry of Health [33]. During 2019, there were 5,811,240 patient visits to OPDs in teaching hospitals of Sri Lanka [33]. Sometimes medical students undergo training at a few other hospitals which are not in these official lists. Rehabilitation Hospital-Ragama has no OPD or ETU. Fig 2 illustrates that most of them are located in densely populated areas particularly in the South-West of the country where most of the population inhabits.

### Objectives

Our main objective was to get a basic idea of indoor $CO_2$ levels at OPDs and ETUs of teaching hospitals of Sri Lanka during morning rush hours as a proxy to estimate the risk of respiratory infection transmission. Our secondary objective was to get a basic idea of $PM_{10}$, $PM_{2.5}$,total particles, air temperature, humidity, total VOC (TVOC) and formaldehyde levels in the air of the same places at the same time.

### Data

We visited each hospital during November 28, 2022 to May 10, 2023 period and took measurements 8.15a.m.-12noon. Complying with the recommendations given in the national IAQ guidelines, we took measurements in three days in each hospital [20]. Total of three measurements per day from OPD, ETU and outdoors of each hospital. Each of those measurement consisted of $CO_2$, air temperature, humidity, $PM_{10}$, $PM_{2.5}$, total particles, VOC and formaldehyde levels. There were 63 measurements per one round covering all 21 hospitals and a total of

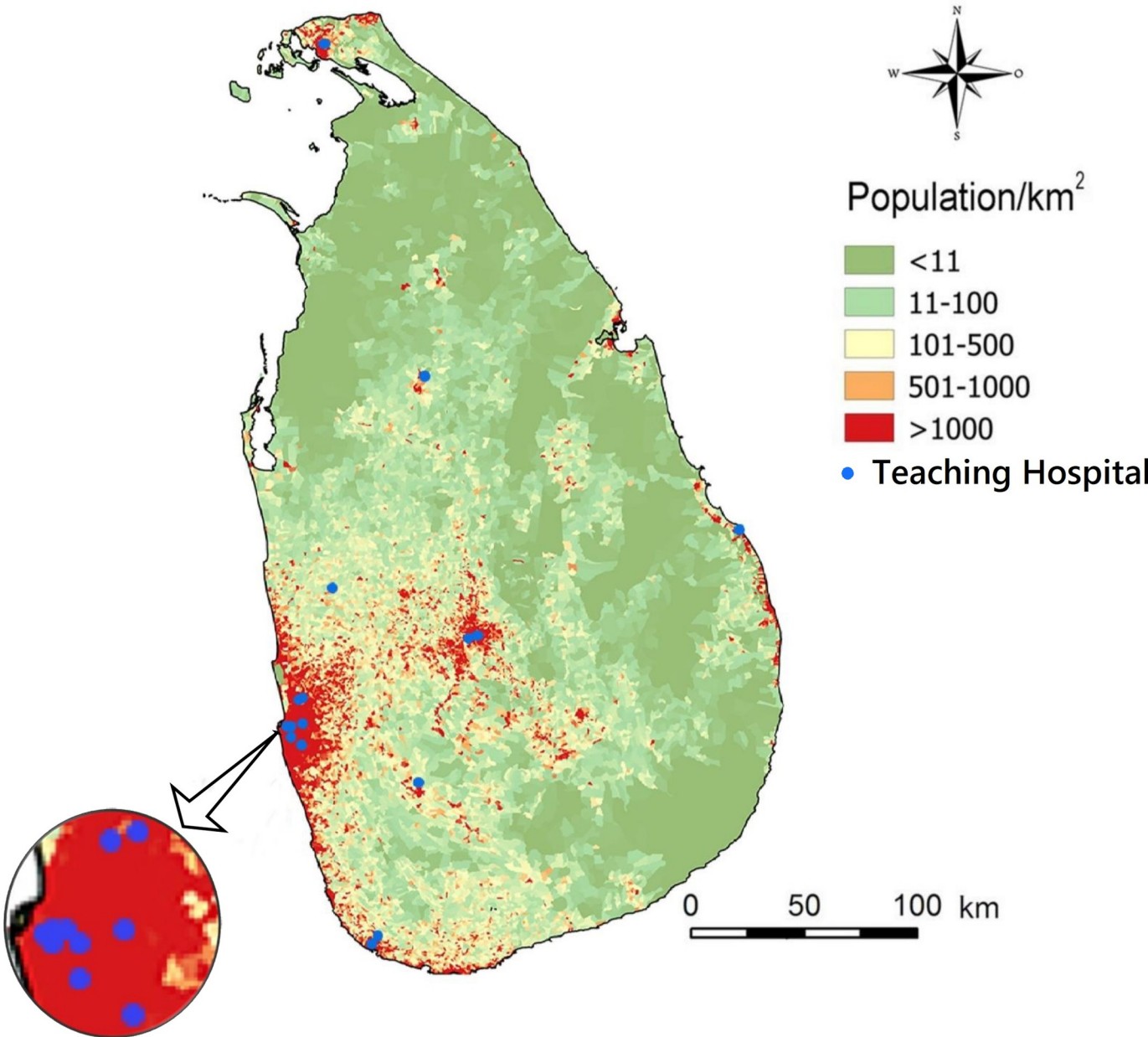

**Fig 2. The locations of teaching hospitals of Sri Lanka participated in this study are marked as blue dots in the map.** The background shows population density in color codes. Due to the proximity of some hospitals to each other, some blue dots overlap at this scale. We adopted Fig 1 of https://doi.org/10.1371/journal.pntd.0004813 published under CC BY4.0. [34] as the base layer of this map.

189 measurements for the entire study. There was no missing data in this data set. We maintained at least a 10 days gap between two measurements in each hospital. We used AZ 77535 (AZ Instrument Corp. Taiwan) to measure $CO_2$, air temperature and humidity levels and Temtop LKC-1000S+ Second Generation Professional (Elitech Corp. U.S.A.) to measure $PM_{10}$, $PM_{2.5}$, total particles, TVOC and formaldehyde levels. The measuring range of the AZ

77535 for $CO_2$ was 0-9999ppm, but when $>5001$ppm it was out of scale. The accuracy of the machine was $\pm30$ppm$\pm5\%$ when $CO_2$ level was $<5000$ppm. Measuring range of the Temtop LKC-1000S+ for both $PM_{2.5}$ and $PM_{10}$ was 0–999 μg/m$^3$ with a resolution of 0.1 μg/m$^3$. And the measuring range for formaldehyde and total VOC was 0–5 mg/m$^3$ with a resolution of 0.01 mg/m$^3$. We took continuous 15minute short-term exposure limit measurements at a time (STEL). Which means the average value of the 15 minutes as one measurement. The meters were placed 1-2m (usually 1.5 m) above the floor, well away from fresh air inlets (windows) and at least 1m away from people [8]. Before taking indoor measurements, we took measurements outdoors, at the same height, at least 1m away from people. We obtained two sets of brand-new machines calibrated and quality checked at the factory. We used one set for taking measurements. After each round (63 measurements) we checked for accuracy/reliability of machines by keeping machines used for data gathering and reference machines set 2-3cm apart and simultaneously measuring air quality parameters in an outdoor open space. The measurements were in good agreement within ± 10%.

## Exclusion of certain days from our study

On certain days (i.e.: after a crash of a crowded bus), OPDs and ETUs are visited by above average numbers of patients and their loved ones. On certain other days (i.e.: stormy days, public holidays) below average numbers of visitors turn up. To mitigate the effects of such extremes affecting our data, we recorded data on three days at each hospital and took the average values and excluded Sundays and public holidays.

## Analysis

We calculated the averages of $CO_2$, $PM_{10}$, $PM_{2.5}$, total particle, formaldehyde and VOC levels for hospital outdoors, OPDs and ETUs by taking the mean and median of measurements of the three data sets (63 data points for each parameter). We calculated the average differences between outdoor $CO_2$ concentration and the OPD and ETU $CO_2$ concentrations of each hospital to get an idea of ventilation. We calculated the mean and median indoor temperature and RH values. We calculated the correlation between certain outdoor AQI parameters ($PM_{2.5}$, $PM_{10}$) and the corresponding indoor AQI levels of OPDs and ETUs as well.

We looked for any statistically significant differences in between our $CO_2$ data sets by performing the paired t-test. Analyses were done using Microsoft Excel 2010 software and IBM SPSS 23.0 software. SPSS 23.0 software results are declared as (SPSS 23.0 results) and the rest are Excel 2010 results.

## Ethics statement

This study was approved by the ethical review committee of the Teaching Hospital-Peradeniya (THP/PLANNING/ERC/19/2021).

## Results

Table 1. Summarizes the average IAQ parameter values.

Guidelines for IAQ in Sri Lanka 2022 by the Ministry of Environment-Sri Lanka [20] have adopted ceilings set by authoritative institutions of other countries as depicted in the table.

S1 Data gives detailed IAQ values recorded at each hospital each day.

Fig 3 illustrates the average $CO_2$ concentrations of OPDs and ETUs of each hospital.

Paired two-tailed t-test P values between our first and second round OPD $CO_2$ data set and the first and third round data sets were $>0.05$, indicating there was no significant difference

**Table 1. Averages of IAQ parameters.**

| Parameter | Outdoors Mean (Median) | OPDs Mean (Median) | ETUs Mean (Median) | National Ceiling | International Ceiling |
|---|---|---|---|---|---|
| Avg.STEL $CO_2$ concentration(ppm) | 514 (510) | 749 (673) | 795 (703) | 5000(8hr)-OSHA PEL | 5000(8hr)-(OSHA PEL) <1000(Germany, Canada-ref10) |
| $CO_2$ concen.above outdoor level(ppm) | - | 235(152) | 283(197) | Outdoor conc.+700 (ASHRAE) | Outdoor conc.+700 (ASHRAE) |
| MIN $CO_2$ concen.(ppm) | 499 (495) | 673 (618) | 709 (663) | | |
| MAX $CO_2$ concen.(ppm) | 570 (561) | 827 (751) | 863 (774) | | |
| Temperature (Celsius) | 29.6 (29.5) | 28.6 (28.5) | 26.9 (26.9) | 20–26.1 range (ASHRAE) | |
| RH (%) | 65.2 (66) | 70.4 (71.6) | 67.2 (68.2) | 30–65 range (ASHRAE) | 40-60range (ASHRAE)ref30 |
| $PM_{2.5}$ (µg/m$^3$) | 28.7 (24) | 32 (28.9) | 25.6 (19.7) | <50 (24 hr) | <15 (24hr)-(WHO) |
| $PM_{10}$ (µg/m$^3$) | 49.4(40.9) | 55.5(47.7) | 47.9 (39) | <100 (24hr) | <45 (24hr)-(WHO) |
| Total Particulate Matter per liter | 4314(3632) | 4850 (4349) | 4183(3433) | 15µg/m$^3$ (OSHA) | |
| Formaldehyde (mg/m$^3$) | 0.03 (0.03) | 0.06 (0.04) | 0.12 (0.08) | 0.016–0.1(15min) (NIOSH) | 0.75-2(15min) (OSHA) |
| TVOC (mg/m$^3$) | 0.13(0.12) | 0.31(0.19) | 0.50 (0.38) | 1 (8hr) | |

Avg.STEL = Average Short-term Exposure Limit, OSHA PEL = Occupational Safety and Health Administration (of USA) permissible exposure limits, ASHRAE = The American Society of Heating, Refrigerating and Air-Conditioning Engineers, WHO = World Health Organization, NIOSH = National Institute for Occupational Safety and Health (of USA). Reference 10: Health Canada and German Umweltbundesamt ceilings, MIN = minimum, MAX = maximum

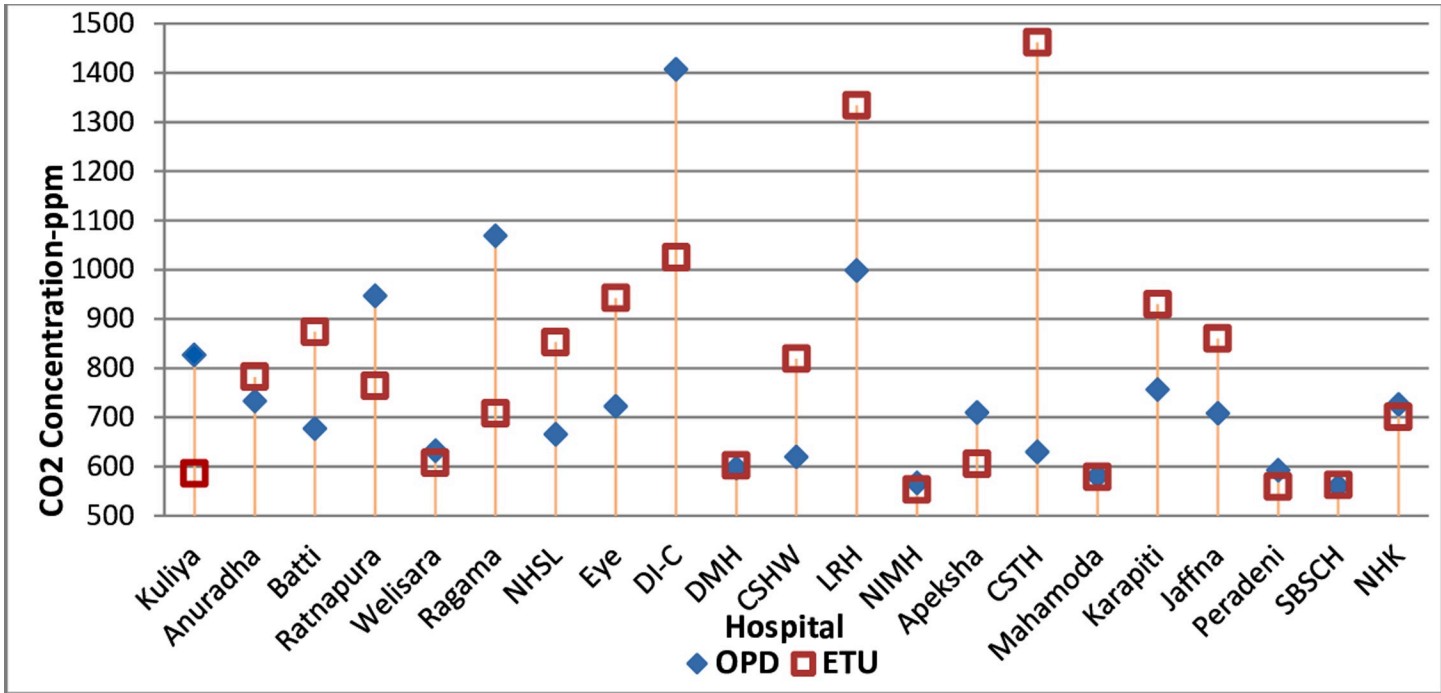

**Fig 3. Average CO2 concentrations of OPDs and ETUs of each hospital.** This figure was created by the first author using Microsoft Office 2010 software. National Hospital Colombo (NHSL), National Hospital Kandy (NHK), Teaching Hospital (TH) Peradeniya (Peradeni), TH Karapitiya (Karapiti), Castle Street Hospital for Women (CSHW), De Soysa Hospital for Women (DMH), TH Mahamodara (Mahamoda), TH Anuradhapura (Anuradha), TH Colombo North Ragama (Ragama), TH Batticaloa (Batti), TH Jaffna (Jaffna), Lady Ridgeway Hospital for Children (LRH), Sirimavo Bandaranaike Specialized Childrens Hospital (SBSCH), TH Colombo South Kalubowila (CSTH), TH Kuliyapitiya (Kuliya), TH Ratnapura (Ratnapura), National Eye Hospital-Colombo (Eye), TH (Chest hospital)-Welisara (Welisara), Rehabilitation Hospital-Ragama (RHR), Dental Institute-Colombo (DI-C), National Institute of Mental Health-Angoda (NIMH), National Cancer Institute Maharagama (Apeksha).

between the first versus second and third round data sets. We did Wilcoxon signed-ranks test (SPSS 23.0) and confirmed that there was no significant difference between the first versus second and third round data round sets. Similarly, paired two-tailed t-test P values plus Wilcoxon signed-ranks test (SPSS 23.0) demonstrated no significant difference between the first versus second and third round data sets of ETUs as well. The difference between the averages of maximum and minimum $CO_2$ levels as a percentage of the average mean $CO_2$ level of OPDs was 21%. The difference between the averages of maximum and minimum $CO_2$ levels as a percentage of the average mean $CO_2$ level of ETUs was 19%. TH Ragama OPD on one day and OPD of the Dental Institute on all three days had $CO_2$ levels 700ppm higher than the corresponding outdoor levels. The ETU of CSTH on all three days and the ETU of LRH on two days had $CO_2$ levels 700ppm higher than the corresponding outdoor levels.

Pearson's correlation coefficients(r) between outdoor $PM_{2.5}$ and $PM_{10}$ levels and OPD levels, respectively, were 0.86 and 0.87(very strongly correlated) [35]. The Pearson's r between outdoor $PM_{2.5}$ and $PM_{10}$ levels and indoor levels in 10 ETUs without air-conditioning, respectively, were 0.63 and 0.74 (moderately correlated). The Pearson's r between outdoor $PM_{2.5}$ and $PM_{10}$ levels and indoor levels in 11 ETUs with air-conditioning, respectively, were 0.40 and 0.66 (fair to moderately correlated) .However, on one day the air-conditioner of the Lady Ridgeway Hospital was switched off. On one day indoor $PM_{2.5}$ and $PM_{10}$ levels of Chest hospital- Welisara were unusually high. Average $PM_{2.5}$ and $PM_{10}$ levels of ETUs without air-conditioning, respectively, were 24.3 μg/m$^3$ and 44.5 μg/m$^3$. Average $PM_{2.5}$ and $PM_{10}$ levels of ETUs with air-conditioning, respectively, were 26.9 μg/m$^3$ and 51.0 μg/m$^3$. The average value of 39 PM2.5 data points obtained from OPDs from the end of November 2022 to the end of February 2023 was 39.1 μg/m3 and the corresponding outdoor level was 33.7 μg/m3. The average value of PM2.5 data obtained from 11 OPDs in May 2023 was 15.3 μg/m3.

Average indoor temperature of 18 OPDs in the low country area was 28.9˚C. Average indoor temperature of three OPDs situated in the hill country about 500m above mean sea level was 27.3$^0$ C. Average indoor temperatures of 7 naturally ventilated ETUs in the low country was 27.9$^0$ C. Average indoor temperatures of 11 air-conditioned ETUs, all in the low country, was 26.2$^0$ C. Average indoor humidity of the 11air-conditioned ETUs was 65.4%. Average indoor RH% of 10 naturally ventilated ETUs was 69.2%.

## Discussion

Our results indicate that during the morning peak hours, the average indoor $CO_2$ levels of the OPDs and ETUs of all of Sri Lankan teaching hospitals were below the ceiling mentioned in the national IAQ guidelines [20]. However, two OPDs and another two ETUs had $CO_2$ levels 700ppm higher than corresponding outdoor levels (another ceiling mentioned in national guidelines). Indoor $PM_{10}$, $PM_{2.5}$ were generally below the national IAQ ceilings but were well above the WHO's ceilings [20]. Average formaldehyde levels of ETUs were slightly higher than the ceiling given in national IAQ guidelines but lower than some international ceilings [20]. On December 12, 2022 $PM_{2.5}$ and $PM_{10}$ levels of both OPD and ETU of Chest hospital- Welisara were very high. Corresponding outdoor levels were also high but the gap between indoor and outdoor levels was large. Therefore we think high outdoor levels plus additional reason/s not noticed by us contributed to these very high indoor particulate matter levels.

To our knowledge, this is the first nationwide study that assesses $CO_2$ levels and other IAQ parameters at all teaching hospital OPDs and ETU/A&Es of any country.

## Discussion of $CO_2$ levels

The first Sri Lankan national IAQ guidelines published recently as part of the "clean air 2025 action plan" do not directly give a ceiling for indoor $CO_2$ for public spaces but mention the upper limits set by some foreign institutions (800-5000ppm) [20]. Those guidelines give special attention to the OSHA (USA) 5000ppm ceiling for eight hours and to the 700ppm plus ambient level ceiling set by ASHRAE (USA) [20]. These upper limits were not primarily targeted to reduce COVID-19 /respiratory infection transmission risk. Different public health and professional institutions of North America and Europe recommend different ceilings of long-term indoor $CO_2$ concentrations, 700-1500ppm to reduce COVID-19 transmission risk [8, 10]. There are no national indoor $CO_2$ level ceilings in many low and middle income countries to our knowledge.

Average outdoor $CO_2$ level of 514 ppm in hospital premises is higher than the global average value [7]. We measured outdoor levels 5-10m away from the main entrances of hospitals. OPDs and ETUs are usually situated close to the main entrances of hospitals. All these hospitals are in highly populated areas as shown in Fig 3, vehicular and people movements are high near hospital entrances and usually there are nearby major roads. Those factors explain the high outdoor $CO_2$ levels. For comparison in Shanghai, China, the average value of $CO_2$ concentration on an auto road was $550 \pm 20$ ppm [36].

WHO guidelines 2009 strongly recommended adequate ventilation in healthcare facilities in all patient care areas as a necessity [6]. It further recommends a minimum ventilation rate of 60 l/s/patient for OPDs [6]. However, some studies indicate even lower levels of ventilation as good [9, 37].

Maintaining indoor $CO_2$ concentrations <1000ppm is recommended by many authoritative institutions and was demonstrated to be containing transmission of tuberculosis by a study conducted at a university in Taiwan [8, 10, 37, 38]. At universities, infectious cases are sparse and most people are healthy young adults. At hospital OPDs and ETUs, infectious patients with respiratory infections are more concentrated than in most other public places like universities. The outdoor airflow rate per person necessary to prevent nosocomial infection may increase linearly with the number of infectious patients in the premises [9]. The fraction of people with weak immune systems due to other diseases is also high in OPDs and ETUs. They can contract an infection with even lesser loads of a less virulent strain of the same virus and spread it to others [39]. Hence, $CO_2$ levels considered not significantly risky in places inhabited by healthy people under normal conditions where patients with respiratory infections are sparse, might be risky in OPDs and ETUs, especially whilst respiratory infection incidence is high in the community (COVID-19 pandemic). Further studies are necessary to determine acceptable $CO_2$ ceilings for OPDs and ETUs of Sri Lankan/ tropical developing world public hospitals. Considering all, we believe maintaining indoor $CO_2$ levels practically possible closest to outdoor air levels in hospital settings would be useful to reduce hospital acquired respiratory infections. Already some local government agencies of certain countries like the National Collaborating Centre for Environmental Health of Canada recommend keeping $CO_2$ levels of indoor air as close to outdoor air during pandemic conditions [8]. Washington State Department of Health of the USA, as a part of their pandemic management strategies in 2021 recommended restaurants to monitor $CO_2$ levels and relocate patrons if $CO_2$ levels exceed 450ppm for 15 minutes [40].

During our visit, OPDs had many more people than ETUs. Nevertheless, in 10/21 hospitals average $CO_2$ concentrations were higher in ETUs. Smaller volume of space in ETUs and OPD walls usually having more area open to the outside may be the reason for this. We believe that

this is a good explanation why $PM_{2.5}$ and $PM_{10}$ concentrations of OPDs were more strongly correlated with outdoor levels.

All OPDs we studied have natural ventilation. Enhancing indoor ventilation can be done without a cost by keeping existing windows open whenever possible, especially during rush hours (but covered by meshes to keep off intruders from mosquitoes to monkeys) and installing exhaust fans at appropriate places. Some OPDs have a limited number of fans. Nevertheless, there is a glitch. Outdoor $PM_{2.5}$ levels in Sri Lanka and in neighboring South Asian countries are higher than WHO ceilings. The situation in Sri Lanka is better than her neighbors [20, 41, 42]. Hence, increasing outdoor air flowing in without filtration is also unhealthy. Sri Lanka is undergoing the worst economic crisis after gaining independence which reached its high point in mid-2022 [43]. Thus, finding finances for installing and maintaining ventilation systems with filtration would be difficult, although important from the health point of view [27, 30]. A recent study done at a fever clinic and A&E of a modern teaching hospital in China shows that $CO_2$ dilution is poor in mechanically ventilated spaces compared to naturally ventilated spaces [9].Thus future systems to improve OPD and ETU ventilation shall be meticulously planned to achieve the objective.

We propose other feasible complementary measures to reduce the risk of respiratory infection transmission in OPDs and ETUs. One of the essential conditions for respiratory infection transmission is that the pathogen concentration around the patient (source) must be sufficiently high to infect another person either due to high numbers exhaled by the patient or due to poor indoor ventilation [6]. Therefore, giving a free surgical mask to every patient with respiratory symptoms with fever and asking to wear it when entering OPDs/ETUs even after the current pandemic to restrict the release of pathogens, efforts to reduce patients' duration of stay at OPDs to reduce the duration of risky exposures are worthy and feasible. Assigning one dedicated OPD doctor on a roster basis to attend patients with respiratory symptoms and separation of those patients from the rest by allocation of a separate area is another option.

There are many factors other than ventilation ($CO_2$ concentration) that determine the risk of transmission of respiratory infections in indoor spaces. Fomites also play a key role in the transmission of those infections [6, 8]. Thus, $CO_2$ concentration is not linearly correlated to the risk of respiratory infections [6, 8]. Nonetheless, the placement of a $CO_2$ concentration monitoring mechanisms at OPDs and ETUs of major hospitals to warn the staff (rapidly rising or >1000ppm concentrations) enables them to take actions such as fully opening existing windows, switching on any exhaust fans may be helpful to reduce the risk of nosocomial infections due to poor ventilation.

We have previously described adverse health effects of high indoor $CO_2$ levels other than increased respiratory infection transmission. Those also indicate that keeping the indoor $CO_2$ level closer to outdoor levels is good for the health of occupants. However, one review article concluded that it is difficult to accurately correlate $CO_2$ levels <5000 ppm with any adverse health effects due to $CO_2$ per se [38]. It further states that $CO_2$ levels <1000 ppm represent good indoor air quality and <1500 ppm are acceptable for the general population [38]. With recent evidence, the ceilings of residential indoor $CO_2$ levels tend to get reduced. For example, Health Canada (responsible for the national health policy of Canada) decreased the ceiling from <3500ppm in 1987 to <1000ppm in 2021 [26].

There are recent success stories of controlling respiratory infections by improved ventilation in the literature. One example is that the aforesaid university in Taiwan managed to control the transmission of tuberculosis very effectively by improving ventilation coupled with $CO_2$ monitoring [37].

$CO_2$ levels of OPDs of Dental institute Colombo which is one of the two premier dental hospitals of the country, were >700ppm higher than outdoor levels in all three days

individually and the average value was 1407ppm. Dental surgeons work close to the open pharynx and airway of patients, most dental OPD procedures generate aerosols [44]. Hence, the risk of contracting nosocomial respiratory infections is especially high at the Dental Institute. Average $CO_2$ level of OPDs of Lady Ridgeway hospital, Sri Lanka's largest children's hospital, was 998ppm and was >700ppm higher than outdoor levels on two days. We explain those two as a result of inadequate ventilation in relation to the population inside. Children are more susceptible to adverse health effects of higher $CO_2$ levels (poor ventilation) [26]. $CO_2$ levels in the OPD of Ragama Teaching hospital also were high. Therefore, the improvement of ventilation in those OPDs deserves priority. The same can be said about $CO_2$ level of ETUs of Lady Ridgeway hospital and Kalubowila hospital (CSTH).

## Discussion of other IAQ parameters

Average $PM_{2.5}$ and $PM_{10}$ levels of both outdoor and indoor air were higher than the current WHO recommended levels. However, they were lower than national ceilings. There is a diurnal variation of $PM_{2.5}$ levels and we took measurements during the peak period of this variation [41]. During November-February, North East monsoon (Indian winter monsoon) winds bring relatively highly polluted air from the mainland of the subcontinent to Sri Lanka [41]. Several times during our period of study, the National Building Research Organisation (a state agency monitoring air quality in Sri Lanka) issued a warning of the arrival of such polluted air masses and we give one example as a reference [45]. The average value of 39 $PM_{2.5}$ data points obtained from OPDs from the end of November 2022 to the end of February 2023 was 39.1 $\mu g/m^3$ and the corresponding outdoor level was 33.7 $\mu g/m^3$. In comparison, the average value of $PM_{2.5}$ data obtained from 11 OPDs in May 2023 was 15.3 $\mu g/m^3$. The corresponding outdoor level was 15.0 $\mu g/m^3$. May to September is the South West monsoon (Indian summer monsoon) season when the wind blows from the Indian Ocean in the opposite direction. Further studies would be helpful to clearly understand how polluted air from other countries and monsoon winds influence indoor PM levels in Sri Lanka.

All OPDs were naturally ventilated and 10 out of 21 ETUs were naturally ventilated. $PM_{2.5}$ and $PM_{10}$ levels of those OPDs and ETUs were moderately to very strongly correlated with the respective outdoor levels [35]. Thus, the high PM level in the air (of those) can be explained by high outdoor levels as in some past studies [23, 31]. Even in 11/21 ETUs with air-conditioning, $PM_{2.5}$ and $PM_{10}$ levels were moderately correlated with outdoor levels. Past authors also have stated that indoor air pollution as a direct result of outdoor air pollution [46]. We expected average particulate matter level of air-conditioned ETUs to be lower than that of naturally ventilated ETUs. Nonetheless, average levels of both $PM_{2.5}$ and $PM_{10}$ were slightly higher in air-conditioned ETUs. We think further studies are needed to find reasons for this.

A systematic review and meta-analysis of hospital IAQ studies illustrated that concentrations of airborne microbes measured as aerobic colony count were correlated with $CO_2$ and $PM_5$ levels [47]. This indicates that improving those IAQ parameters by improving ventilation and air filtration may be helpful to reduce nosocomial infections by reducing concentration of airborne microbes.

Considering the available good evidence of the numerous and serious and long-lasting adverse health effects of PM, considering the health of millions of patients who seek care in them annually and the health of the staff, and considering the health benefits of reducing the risk of nosocomial respiratory infections, taking the initiative to install standard air filtration and central ventilation systems at OPDs and ETUs of teaching hospitals at earliest possible time would benefit Sri Lankan public [23, 26, 31, 33, 42]. Baseline air changes rate per hour of those ventilation systems can be estimated considering the volume of the building, WHO

recommended ventilation rate per occupant and average number of occupants [6]. The system can be coupled with a $CO_2$ monitor to do fine tuning of the ventilation depending on $CO_2$ levels to make the process efficient. There are other cheaper bridging solutions to be considered such as the use of portable home air purifiers when PM levels are unusually high coupled with PM level monitoring [48].

Reports about total particulate matter in public indoor spaces are sparse. We measured total particle count, and the national guidelines mention weight. Thus, a comparison of our total particulate matter results is difficult.

Indoor PM and VOC levels usually rise after hospital cleaning/disinfection activities [31]. Therapeutic procedures like nebulization also release PM to the air [31]. Nebulization is a common procedure done in ETUs. People and equipment regularly move in and out of and inside OPDs and ETUs on foot, wheelchairs and in trolleys and that also leads to settled PM to get re-suspended in air and to prevent PM settling down. We believe that higher indoor PM and VOC levels compared to the outdoor levels we observed in many hospitals can be partially explained by those [31]. There may be other reasons as well.

A systematic review on IAQ in inpatient environments citing ASHRAE gives RH of 40%-60% as the recommended range [31]. Generally in most parts of Sri Lanka RH is high and the lowest RH levels are recorded during December-February. The average RH of OPDs plus naturally ventilated ETUs was 70%.The same review states temperature shall be >21–24°C [31]. All OPDs and ETUs of our study comply with this requirement. ASHRAE recommends an upper indoor temperature limit of 27°C for summer which may be applicable to a tropical country like Sri Lanka [49]. All OPDs (naturally ventilated) and seven naturally ventilated ETUs had average temperatures above this ceiling. Eleven air-conditioned ETUs had average temperatures 23.5–27.7°C. Hence, naturally ventilated OPDs and ETUs are hotter and humid than commonly used standards. However, further studies are necessary to get a better idea of the situation.

## Comparison of the results of similar past studies with our results

Indoor PM levels were exceeding ceiling by 2–3.5 folds and correlated with outdoor levels and VOC and formaldehyde levels were higher in indoor air than outdoors of hospitals in Lebanon like in Sri Lankan hospitals [23]. Indoor air formaldehyde level of naturally ventilated roadside dwellings of Colombo Sri Lanka was 3.3 μg/m³ and total carbonyl (VOC) levels was 13.6 to 18.6 μg/m³ in a past study [50]. In our study the corresponding median values of OPDs, respectively were 40 and 80 μg/m³. The corresponding median values of ETUs, respectively, were190 and 380 μg/m³. Much higher VOC levels inside OPDs and ETUs were may be mainly due to chemicals used for disinfection and cleaning. Further studies may clarify this. Authors of reference 52 have identified emissions from traffic as the main source of indoor carbonyls (VOC) [50]. A study done at three large hospitals in Bangladesh has demonstrated higher indoor $CO_2$, $PM_{10}$, $PM_{2.5}$ and VOC levels than our average findings [51]. Their indoor PM levels were lower than outdoor levels. Like us, they also found higher VOC levels indoors [51]. Other researchers have demonstrated that the levels of several VOCs are on average 2 to 5 times higher indoors than outdoors [52]. A study conducted in three hospitals in Northern India demonstrated lower indoor $PM_{2.5}$ levels than outdoors. Their indoor $PM_{2.5}$ levels were correlated with outdoor levels like in our study and both those levels were higher than in our results [53]. IAQ of six hospitals in Lahore, Pakistan was studied. Their outdoor and indoor PM levels were also higher than corresponding our results [54].Those researchers found that central air-conditioning improves IAQ by reducing $PM_{2.5}$ and $PM_{10}$ and $CO_2$ levels as compared to non-central air-conditioning [55]. Considering aforesaid limited literature, we can

say that IAQ of Sri Lankan teaching hospital OPDs and ETUs is generally better than IAQ of hospitals of neighboring South Asian countries. However more studies are necessary to get a clear picture.

An IAQ study was done in five OPDs of a hospital in Thailand. Those researchers concluded that average $CO_2$ levels >1000ppm and high airborne Staphylococcus bacterial concentration they detected indicate poor ventilation with overcrowding and unhealthy IAQ in those OPDs [55], A study done in a Saudi Arabian hospital showed lower indoor $PM_{10}$ levels than outdoor $PM_{10}$ levels and both levels were much higher than our levels [32]. They also established higher VOC levels indoors. They detected more particulate matter in air in sections of the hospital with more human activity [32].

## Limitations

Indoor CO2 concentration has been used as a proxy of the degree of ventilation and, by extension, as an indicator of the risk of contracting respiratory infections in that space [5, 8–11, 37]. Based on available literature that we mentioned in the introduction and in discussion we are convinced that the concept is valid [5, 8–11, 37, 40, 48]. Thus we did not attempt to prove that concept again in this study. We assumed that concept was true and conducted the study although the concept is not universally accepted. We took continuous measurements for 15 minutes duration at one site per day for three days. Taking measurements for a longer duration representing 12 months of the year may give a better idea of the situation. Nonetheless, 15 minutes or longer exposure is considered as a high risk of contracting COVID-19(respiratory infection) by the Sri Lankan Ministry of Health and international agencies like the WHO [56]. We took three measurements at a hospital one after another starting from outdoors. Minor changes of IAQ parameters would have happened after we took one measurement. Nevertheless, we could avoid potential biases associated with employing three equipment sets with three observers simultaneously. We think our study is an eye opener for the Sri Lankan medical community and provides a good basic idea (foundation) of IAQ and the risk of respiratory infections on which future research may be conducted. The difference between average minimum and average maximum levels of $CO_2$ was <25% of the average mean $CO_2$ levels. We took three measurements from one location. That comply with the requirements for similar studies stipulated in our national IAQ guidelines [20]. Moreover, there was no statistically significant difference between our three $CO_2$ data sets of OPDs and ETUs (our main focus).

## Conclusions

Indoor $CO_2$ levels of the OPDs and ETUs of Sri Lankan teaching hospitals are generally below the ceilings mentioned in the national IAQ guidelines [20]. However, they are above the recent ceilings set by authorities of some developed countries [8, 10, 26, 37, 38]. Considering the concentration of infectious patients and other patients with poor immunity in OPDs and ETUs, reducing $CO_2$ levels by improving ventilation complemented with other infection preventive measures would be helpful to reduce the risk of nosocomial respiratory infections. Indoor $PM_{10}$, $PM_{2.5}$ levels in OPDs and ETUs are generally below the national IAQ ceilings but above the WHO's ceilings [20]. Further studies may clarify how monsoon winds modulate indoor $PM_{10}$, $PM_{2.5}$ levels in Sri Lanka. VOC and formaldehyde levels are generally below the ceilings mentioned in the national IAQ guidelines [20]. Installation and regular maintenance of appropriate air filtration and ventilation systems at OPDs and ETUs of teaching hospitals would contribute to improve indoor PM and VOC levels and reduce the risk of nosocomial respiratory infections. There is a wide range (700-15000ppm) in the upper ceilings of indoor $CO_2$ set by various institutions in developed nations for various durations of exposure to mitigate risk

of respiratory infection transmission or to mitigate adverse health effects of high $CO_2$ levels per se [8, 10, 20, 26]. Further studies shall determine such ceilings appropriate for OPDs and ETUs of large hospitals of Sri Lanka. Limited published evidence and our results show that IAQ of OPDs and ETUs of Sri Lankan teaching hospitals is generally better than IAQ of hospitals of neighboring South Asian countries [52–54].

## Supporting information

**S1 Data. Hospital air quality dataset.**
(XLSX)

## Acknowledgments

The authors sincerely thank all who cooperated in our data gathering and who helped us to publish this paper.

## Author Contributions

**Conceptualization:** N.D.B. Ehelepola.

**Data curation:** N.D.B. Ehelepola, H.M. Arjuna Thilakarathna.

**Formal analysis:** N.D.B. Ehelepola.

**Funding acquisition:** N.D.B. Ehelepola.

**Methodology:** N.D.B. Ehelepola.

**Project administration:** N.D.B. Ehelepola.

**Resources:** N.D.B. Ehelepola.

**Supervision:** H.M. Arjuna Thilakarathna.

**Visualization:** N.D.B. Ehelepola.

**Writing – original draft:** N.D.B. Ehelepola.

**Writing – review & editing:** N.D.B. Ehelepola, H.M. Arjuna Thilakarathna.

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
