## [Decision Letter · Decision Letter 0]

5 Nov 2023

PGPH-D-23-01122

Respiratory Infection Transmission Risk and Indoor Air Quality at Outpatient Departments and Emergency Treatment Units of Sri Lankan Teaching Hospitals.

Dear Dr. Ehelepola,

Thank you for submitting your manuscript to PLOS Global Public Health. After careful consideration, we feel that it has merit but does not fully meet PLOS Global Public Health’s publication criteria as it currently stands. Therefore, we invite you to submit a revised version of the manuscript that addresses the points raised during the review process.

Please see the comments of four reviewers below and in the attachment. All reviewers seem positive about the contributions of the study, and have suggested revisions to strengthen the study.

Please note that reviewer 1's comments about figures being blurry is only a result of the generation of the reviewer PDF, and that the original resolution in which the figures are uploaded is fine, so there is no need to enhance these figures. Please ensure that no figures are copyrighted, and please insert information on how the figures were created, and by whom, in the figure legend, and whether any base maps or stock figures were used.

We look forward to receiving your revised manuscript.

Kind regards,

Hanna Landenmark

Staff Editor

Journal Requirements:

2. Some material included in your submission may be copyrighted. According to PLOS’s copyright policy, authors who use figures or other material (e.g., graphics, clipart, maps) from another author or copyright holder must demonstrate or obtain permission to publish this material under the Creative Commons Attribution 4.0 International (CC BY 4.0) License used by PLOS journals. Please closely review the details of PLOS’s copyright requirements here: PLOS Licenses and Copyright. If you need to request permissions from a copyright holder, you may use PLOS's Copyright Content Permission form.

Potential Copyright Issues:

Figure 3: please (a) provide a direct link to the base layer of the map (i.e., the country or region border shape) and ensure this is also included in the figure legend; and (b) provide a link to the terms of use / license information for the base layer image or shapefile. We cannot publish proprietary or copyrighted maps (e.g. Google Maps, Mapquest) and the terms of use for your map base layer must be compatible with our CC-BY 4.0 license. 

Additional Editor Comments (if provided):

Reviewers' comments:

Reviewer's Responses to Questions

**Comments to the Author**

1. Does this manuscript meet PLOS Global Public Health’s publication criteria? Is the manuscript technically sound, and do the data support the conclusions? The manuscript must describe methodologically and ethically rigorous research with conclusions that are appropriately drawn based on the data presented.

Reviewer #1: Partly

Reviewer #2: Partly

Reviewer #3: Partly

Reviewer #4: Partly

2. Has the statistical analysis been performed appropriately and rigorously?

Reviewer #1: No

Reviewer #2: No

Reviewer #3: No

Reviewer #4: Yes

3. Have the authors made all data underlying the findings in their manuscript fully available (please refer to the Data Availability Statement at the start of the manuscript PDF file)?

Reviewer #1: No

Reviewer #2: Yes

Reviewer #3: Yes

Reviewer #4: Yes

4. Is the manuscript presented in an intelligible fashion and written in standard English?

Reviewer #1: Yes

Reviewer #2: Yes

Reviewer #3: No

Reviewer #4: No

5. Review Comments to the Author

Reviewer #1: Major comments:

While the study focuses on measuring several indexes of indoor air quality (IAQ), the authors attempt to link IAQ with the risk of respiratory infection. The Introduction and Discussion sections heavily emphasize the association between IAQ and "nosocomial infection" or "COVID-19." While highlighting these links is understandable to some extent, the data and analyses presented in this study do not actually support such claims. For example, in line 276, the authors stress that "this is the first nationwide study that assesses the risk of respiratory infection transmission using CO2 levels…," which seems inappropriate unless the authors also evaluated the presence of viruses or bacteria in the indoor air samples or evaluate the association between IAQ and risk of respiratory disease infection.

Table 1 and Figure 3 makes it difficult to understand the overall statistical analysis. It is not clear what was the analysis unit for this study. Are the observations for 15 minutes in a single hospital considered as a single unit, or the real time (seconds or minutes averaged) values for each hospital are used for analysis? How was the data averaged? Were there any missing values? Moreover, it is essential to clarify if the sample size was large enough to conduct t-tests. These issues should be carefully addressed in the statistical analysis section, results, and tables to enhance reader understanding.

The Figures 1 and 2 is not suitable for publication in scientific journals. The resolution of Figure 3 should be increased, accompanied by a proper explanation. If the authors intend to highlight the difference between outdoor and indoor CO2 levels, Figure 3 should also include outdoor CO2 levels for comparison.

I hope my comments and suggestions will be helpful in refining this manuscript. Please let me know if any further clarification or information is required.

Reviewer #2: Plus Factors- Good study design covering PAN-country; honest mentions of limitations, study is good starting point / a foundation for larger longitudinal / cohort studies.

Lacunas: Had great access to hospital but lack covering patient admission and visit insight data, elaboration on probable pollution sources in premises of identified pollutants, methodology reference or standardization to be added, monitoring instrument calibrations frequency and use indoors-outdoors to be elaborated, line no 150 -bus crash accident- no mention of omission/exclusion in present study, Fig-3 hospital legend to mention, also the hospital density and/or ratio wrt population, ventilation coefficient and air exchange rate should be part of the study as they mentioned the importance and consideration in placing monitors, lack in analysis, correlation, and interpretation of parameters other than CO2, or else remove those parameters, literature review and elaborative discussions in respective sections will have value addition.

Reviewer #3: The article Titled “ Respiratory Infection Transmission Risk and Indoor Air Quality at Outpatient Departments and Emergency Treatment Units of Sri Lankan Teaching Hospitals” is a good attempt to evaluate the air quality in hospitals and medical care facilities. However in its present form it looks a little jumbled up and the authors need to sort out the sectioning to bring out the clarity. Furthermore, the following comments must also be addressed to enhance the quality of the article

Abstract:

1. Define abbreviations

2. Define all the measured parameters.

3. Which particular volatile organic compounds were measured?

4. Why COVID-19 is mentioned as a keyword, although its nowhere mentioned in the abstract.

Introduction

1. There is no need to define CO2 again

2. The introduction is long and disjointed. The authors must make it short and relevant to the theme of the article.

Material and Methods

1. On which basis the hospital selection was done

2. Instrumentation is not adequate

3. The sample size was not adequate.

Results and Discussion

1. Some higher order statistics must be implemented to represent the results

2. Correlate the findings with the previous literature.

3. The health risk assessment of these pollutants must be done.

Reviewer #4: Although, the paper is discussing the indoor CO2 levels which will serve as a proxy for representing the ventilation levels, a simple calculation of Air changes per hour (ACH) in hospitals with manual ventilation would provide a strong foundation for designing efficient ventilation systems.

6. PLOS authors have the option to publish the peer review history of their article (what does this mean?). If published, this will include your full peer review and any attached files.

**Do you want your identity to be public for this peer review?** For information about this choice, including consent withdrawal, please see our Privacy Policy.

Reviewer #1: No

Reviewer #2: No

Reviewer #3: **Yes: **Dr. Tahmeena Khan

Reviewer #4: No

---

## [Decision Letter · Decision Letter 1]

22 Dec 2023

PGPH-D-23-01122R1

Respiratory Infection Transmission Risk and Indoor Air Quality at Outpatient Departments and Emergency Treatment Units of Sri Lankan Teaching Hospitals.

Dear Dr. Ehelepola, 

Thank you for submitting your manuscript to PLOS Global Public Health. After careful consideration, we feel that it has merit but does not fully meet PLOS Global Public Health’s publication criteria as it currently stands. Therefore, we invite you to submit a revised version of the manuscript that addresses the points raised during the review process.

We look forward to receiving your revised manuscript.

Kind regards,

Naveen Puttaswamy, Ph.D

Academic Editor

Journal Requirements:

Additional Editor Comments (if provided):

Reviewers' comments:

Reviewer's Responses to Questions

**Comments to the Author**

1. If the authors have adequately addressed your comments raised in a previous round of review and you feel that this manuscript is now acceptable for publication, you may indicate that here to bypass the “Comments to the Author” section, enter your conflict of interest statement in the “Confidential to Editor” section, and submit your "Accept" recommendation.

Reviewer #1: (No Response)

Reviewer #3: All comments have been addressed

Reviewer #4: All comments have been addressed

2. Does this manuscript meet PLOS Global Public Health’s publication criteria? Is the manuscript technically sound, and do the data support the conclusions? The manuscript must describe methodologically and ethically rigorous research with conclusions that are appropriately drawn based on the data presented.

Reviewer #1: Partly

Reviewer #3: Yes

Reviewer #4: Yes

3. Has the statistical analysis been performed appropriately and rigorously?

Reviewer #1: No

Reviewer #3: Yes

Reviewer #4: Yes

4. Have the authors made all data underlying the findings in their manuscript fully available (please refer to the Data Availability Statement at the start of the manuscript PDF file)?

Reviewer #1: Yes

Reviewer #3: Yes

Reviewer #4: Yes

5. Is the manuscript presented in an intelligible fashion and written in standard English?

Reviewer #1: Yes

Reviewer #3: Yes

Reviewer #4: Yes

6. Review Comments to the Author

Reviewer #1: The authors stated, "not trying to prove the concept of CO2 levels are associated with infection risk estimates in this study," but the title and introduction of the paper still present the possibility of misinterpretation by readers. Therefore, I would like to reiterate the opinion that an overall modification of the term “proxy” and use of “respiratory infection transmission risk” in this study. Additionally, please re-examine the prerequisites for t-test analysis and present the results together.

In the authors' cited review paper 49, there was a significant association between CO2 and airborne microbial concentration. However, by considering the limited number of relevant papers (n=3) for the meta-analysis and the research design and small sample size of individual studies, the evidence supporting the authors' claim that CO2 levels can be used as a proxy for infection appears insufficient. It is recommended to describe this paper as a descriptive study on indoor air pollution levels in hospitals and just briefly indicate the potential associations with various infections and indoor air quality.

Please review the figures once again and ensure that each is self-explanatory.

Reviewer #3: The authors have addressed the comments satisfactorily and therefore the manuscript should be accepted for publication.

Reviewer #4: (No Response)

7. PLOS authors have the option to publish the peer review history of their article (what does this mean?). If published, this will include your full peer review and any attached files.

**Do you want your identity to be public for this peer review?** For information about this choice, including consent withdrawal, please see our Privacy Policy.

Reviewer #1: No

Reviewer #3: **Yes: **Tahmeena Khan

Reviewer #4: No

---

## [Editor Report · Decision Letter 2]

8 Jan 2024

Respiratory Infection Transmission Risk and Indoor Air Quality at Outpatient Departments and Emergency Treatment Units of Sri Lankan Teaching Hospitals.

PGPH-D-23-01122R2

Dear Dr. Ehelepola,

We are pleased to inform you that your manuscript 'Respiratory Infection Transmission Risk and Indoor Air Quality at Outpatient Departments and Emergency Treatment Units of Sri Lankan Teaching Hospitals.' has been provisionally accepted for publication in PLOS Global Public Health.

Best regards,

Naveen Puttaswamy, Ph.D

Academic Editor